# Emotion Recognition in Horses with Convolutional Neural Networks

Luis A. Corujo [1], Emily Kieson [2,*], Timo Schloesser [3] and Peter A. Gloor [1]

1   MIT Center for Collective Intelligence, 245 First Street, Cambridge, MA 02142, USA;
    luis3zc@gmail.com (L.A.C.); pgloor@mit.edu (P.A.G.)
2   MiMer Centre, Björnahusvägen 6, 261 93 Saxtorp, Sweden
3   Department of Information Systems and Information Management, University of Cologne, Pohligstrasse 1,
    50969 Cologne, Germany; timo.schloesser@gmail.com
*   Correspondence: emily@mimercentre.org

**Abstract:** Creating intelligent systems capable of recognizing emotions is a difficult task, especially when looking at emotions in animals. This paper describes the process of designing a "proof of concept" system to recognize emotions in horses. This system is formed by two elements, a detector and a model. The detector is a fast region-based convolutional neural network that detects horses in an image. The model is a convolutional neural network that predicts the emotions of those horses. These two elements were trained with multiple images of horses until they achieved high accuracy in their tasks. In total, 400 images of horses were collected and labeled to train both the detector and the model while 40 were used to test the system. Once the two components were validated, they were combined into a testable system that would detect equine emotions based on established behavioral ethograms indicating emotional affect through the head, neck, ear, muzzle, and eye position. The system showed an accuracy of 80% on the validation set and 65% on the test set, demonstrating that it is possible to predict emotions in animals using autonomous intelligent systems. Such a system has multiple applications including further studies in the growing field of animal emotions as well as in the veterinary field to determine the physical welfare of horses or other livestock.

**Keywords:** convolutional neural networks; horse emotion recognition; horse emotion

## 1. Introduction

There is currently no scientific consensus on defining emotion since an emotion is a subjective mental state associated with the nervous system [1], but there is growing research in emotions in animals related to defining emotions as subjective affect that creates both physiological and behavioral responses [2]. There is extensive research in the human world regarding emotional affects and the effects on behavior and emotional expressions (most of which are based on subjective interviews or standardized questionnaires). As humans, we can gauge emotions based on facial cues, voice tone, posture, and other hints [3]. However, our prediction might be wrong, and a person who appears to be happy may in reality be sad [4,5]. This suggests that observers can perceive emotions from the subject but that the truth of the emotion may be lost in interpretation. Animals experience emotions in a similar way that can be just as difficult to discern. Since Charles Darwin, who was one of the first to write about this topic, many scientists have done research in the field of animal emotions. Jaak Panksepp, one of the pioneers in affective neuroscience, identified seven different emotions that animals can feel [6]. Most of the research has been done in mammals because of their similarity to humans and because many of them produce facial expressions that bear a clear resemblance to the expressions seen in humans [7].

There is also growing research indicating the emotional breadth of animals and the ability to measure it through similar physiological and behavioral measures [2,8]. While the

field of research on dog emotions has continued to flourish, research in agricultural animal emotions is growing at a slower rate. There is, however, a large body of research supporting the use of heart rate variability (HRV) in farm animals and the correlation with cortisol (as an indicator of stress) and behavioral responses as a means of assessing emotional states in domestic livestock [9,10].

Horses are still considered livestock by the United States Department of Agriculture (USDA) and have also been studied for correlations between physiological markers (HRV and cortisol) and behavioral patterns, suggesting that there is a possibility of assessing equine emotional states through behavioral indicators. Studies in stress behaviors, for example, link elevated plasma cortisol levels with increased HRV and specific behavioral patterns, such as elevated head and neck position, widened eyes, variations in ear positions (ranging from forward to indicate alert/attentive to pinned back against the head to indicate higher levels of distress), and increased muscle tension and body movements [11–13]. Furthermore, existing ethograms in equine behaviors of feral and semi-feral horse herds show that behavioral expression in feral herds also supports the connection between behavioral expression, intention, and potential emotional affect when taken in the context of social interactions and communication [14–16].

Correlations between physiological measurements and behavioral parameters have already been used to study the psychological and emotional welfare of horses [17–19] and qualitative measures of behavioral indicators have also been used to assess equine emotional state [20,21]. Furthermore, with the development of the Equine Facial Action Coding System (EquiFACS) [22] and the Equine Pain Face coding system [23], researchers and practitioners have become even more aware of the ability to look at muscles in both the body and face to assess expressions of physical and psychological health, especially related to pain and discomfort. Such facial recognition has been incorporated into machine learning and video coding software to help researchers and practitioners develop better techniques to decipher pain in equids [24–26]. The use of these measures and assessment tools supports the development of standardized methods of assessing behavioral parameters and suggesting emotional affect based on existing studies in behavior, physiological measures, and species-specific ethograms.

Technology already exists to examine emotional expressions in humans. With the improvements in technology during the last few decades, researchers have studied multiple ways of how to recognize emotions in humans using different techniques, such as Markov models, artificial neural networks, and Bayesian networks, among others.

The automation of emotion recognition employs three main approaches:

First, there are knowledge-based techniques that predict emotions based on semantic and syntactic knowledge; statistical methods, which commonly are machine learning algorithms that predict emotions given a big enough data set; and hybrid approaches, which are a combination of the previous two. Many companies have been successful in predicting emotions in humans using these approaches. A good example is the work done by Affectiva [27], a company founded at Massachusetts Institute of Technology (MIT) that uses facial and voice cues to predict emotions.

So, the question arises of whether we can create an intelligent system with the ability to predict emotions of animals based on body and facial expressions. Most of the research conducted on animal emotions has been done in mammals, mainly from a psychological and neuroscience perspective [7,28], with growing work done in looking at equine emotional facial expressions using the EquiFACs and Pain Face coding systems [22,23].

In addition, most of the research focuses on the ability of animals to interpret our emotions and not on how we can interpret theirs. Within the group of those that focus on interpreting the expressions of the animals, most of them do it without using an intelligent or autonomous system. A good example is AnimalFacs, a tool for identifying facial movements in non-human species. The researchers explain how we, as humans, can analyze the facial expressions of different primates, dogs, cats, and horses [7,22,29].

There is emerging research done on how to interpret animal emotions using an autonomous or intelligent system. One example is the work done by Laura Niklas and Kim Ferres in predicting dog emotions from images [30]. Another example is the work done at the University of Augsburg on recognizing dog emotions from bark sequences using an autonomous model [31]. Extant facial emotional recognition software in equids focuses primarily on pain expressions [24–26], with more research needed to classify and code a wider range of emotional expressions. Thus, the idea of predicting animal emotions this way is something that has not yet been explored in depth.

This paper explores the possibility of creating an intelligent system capable of predicting the emotion of a horse from its face and neck traits based on existing research and ethograms connecting specific head, neck, eye, nose, and ear positions with specific stress levels and emotional valence [14–16,21,32–34]. In order to do this, two different elements must be created.

First, we need to develop a detector capable of recognizing a horse in an image. More specifically, it needs to detect a region of interest (ROI from now on), which in this case, will be the head and neck of a horse. The task of the detector is a prerequisite for the second part. If the detector does not recognize the appropriate ROI, the second part of the system will not be able to predict the emotion correctly regardless of how well it can accomplish that task.

Secondly, we need a model that, once it receives the detected ROI, is capable of predicting the emotion of the horse. This means this model must be able to detect facial cues and different neck positions and relate them to the appropriate emotion.

Altogether, this should be an intelligent system capable of detecting a horse, analyzing its face and neck features, and predicting its emotion with reasonable accuracy.

## 2. Materials and Methods

### 2.1. Defining a Horse Emotion Tracking Framework

The first step consists of defining the different emotions, independent of interactions with other horses. Physical markers, head, ear, and neck positions are based on research linking behaviors for arousal and physiological stress with only "annoyed", suggesting a negative valence based on previous studies on animal interactions and agonistic behaviors [14–16,21,32–34]. The emotion of "alarmed" was used to indicated heighted arousal based on eye and ear position using existing ethograms with the understanding that the behavioral indicators of this emotion may also indicate heighted vigilance or alertness in addition to alarm [16]. The four emotional markers of "alarmed", "annoyed", "curious", and "relaxed" were chosen due to their representative variations of arousal level and emotional expression within the existing equid ethograms, with "relaxed" representing the lowest level of arousal and "alarmed" representing the highest. The initial photographs were marked and defined by the researchers according to existing ethograms. The term "alarmed" refers to a heightened state of awareness in which the horse demonstrated behaviors indicating higher arousal levels without significant movement of the feet. Horses have a wide range of facial expressions used to express psychological states as well as communicate with other conspecifics, especially with eyes, ears, and head and neck position [15,16,35–38].

Although previous studies of horses have investigated their facial expressions in specific contexts, e.g., pain, until now, there has been no methodology available that documents all the possible facial movements of the horse and provides a way to record all potential facial configurations. This is essential for an objective description of horse facial expressions across a range of contexts that reflect different emotional states. Facial action coding systems (FACSs) provide a systematic methodology of identifying and coding facial expressions on the basis of underlying facial musculature and muscle movement across species [29]. FACSs are anatomically based and document all possible facial movements rather than a configuration of movements associated with a particular situation. Consequently, FACSs can be applied as a tool for a wide range of research questions. The Equine

Facial Action Coding System (EquiFACS) provides a system to measure facial expressions in horses based on musculature and skeletal configurations in equids [22]. On its own, EquiFACS enables researchers to look at distinctive facial movements and changes and, when combined with species-specific knowledge of behaviors, contexts, and behavioral ethograms, creates opportunities to develop connections between these facial configurations and emotional expressions [22]. Portions of EquiFACS focus on the appearance of the sclera (the whites of the eyes), shape of the eye, and tension in the nose, lip, and muzzle, and ear position based on the tension and use of different facial muscles in the horse. EquiFACS also looks at additional musculature and shape changes of the face, lip, nose, eye, and ear positions, which are easily differentiated from one another and appear as markers of behavioral patterns indicative of different levels of arousal and emotional expression in known ethograms of equine behavior [15,16,22,36]. By combining distinctive expressions supported by EquiFACS with known behavioral expressions of equines, we were able to generalize basic emotional expressions of equines based on distinctive changes in nose, lip, eye, and ear positions.

The reliability of others to be able to learn this system (EquiFACS) and consistently code behavioral sequences was high, and this included people with no previous experience of horses [22]. A wide range of facial movements was identified, including many that are also seen in primates and other domestic animals (dogs and cats). EquiFACS provides a method that can be used to document the facial movements associated with different social contexts and thus to address questions relevant to understanding social cognition and comparative psychology, as well as informing current veterinary and animal welfare practices [22,38,39]. These previous results indicate that a combination of head orientation with facial expression, specifically involving both the eyes and ears, is necessary for communicating social attention. The earlier findings emphasize that in order to understand how attention is communicated in non-human animals, it is essential to consider a broad range of cues [22,38,39]. Recent developments in understanding horse facial expression suggest that they have many detailed ways of expressing intention and communication. The use of such detailed facial coding systems like EquiFACS and Pain Face helps develop even better mechanisms for determining levels of stress and pain, especially in close proximity when the nuances of emotional and physical expression need more attention. In order to simplify the emotional coding tool for this experiment and to test a novel system that could be both portable and useful for the layperson, we broke these up into four emotions (Figure 1), which were defined with respect to neck and facial cues that could be judged from a distance:

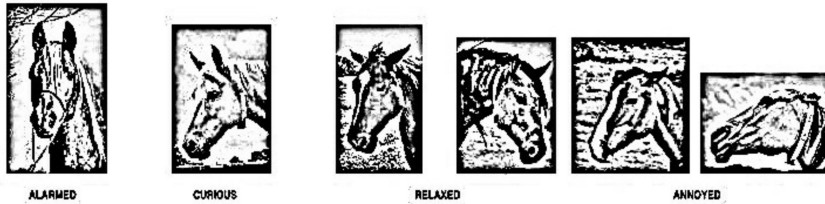

**Figure 1.** Horse emotions.

**Alarmed**

- Eyes: open eyes with little or no sclera
- Ears: stiffly forward
- Nose: open nostrils, usually slightly tense mouth or muzzle
- Neck: above parallel, head higher than back

**Annoyed**

- Eyes: open with perhaps some sclera
- Ears: stiffly back or pinned back, close to the horse's head
- Nose: nostrils slightly closed, tense mouth or muzzle

- Neck: usually parallel or above parallel

**Curious**

- Eyes: open with little or no sclera
- Ears: pointing forward/sides but relaxed
- Nose: open nostrils, relaxed mouth and muzzle
- Neck: usually parallel to ground but may be slightly below or above

**Relaxed**

- Eyes: partially to mostly shut
- Ears: relaxed, opening pointing to the sides
- Nose: relaxed mouth and muzzle
- Neck: approximately parallel or below

Once the emotions were defined, the second step was to collect the data. In this case, a total of 440 images of horses were collected from private sources where the horses were familiar and the context of the photo was known to help guide the coding of expressions where all four criteria were met. There were a total of 110 images per emotion labeled by two of the authors based on the coding system above that was derived from the aforementioned research on behavioral ethograms. Images were of horses at liberty in large paddocks or pastures with no equipment or observed human presence or interactions. These images were split in training and validation sets in order to train the detector and the model.

### 2.2. Detector

To train the detector, 400 images were used as training data, while the 40 left were used as test data. Every picture was rescaled to 200 pixels in height, keeping the original ratio. This was done to facilitate the work of the detector since having fewer pixels requires less computations, and there is no significant information loss when rescaling to this size. All images were labeled. In this case, labeling the images meant to manually highlight the ROI that the detector was supposed to find.

The architecture used for animal detection was the faster region-based convolutional neural network (faster R-CNN) [40], a well-established architecture for object detection. This architecture is composed of three different parts. First, the convolutional layers, which filter the images in order to extract useful features; secondly, the RPN (region proposal network), whose duty is to identify the possible regions where objects (in this case horses) can be located; and finally, a dense neural network that predicts what kind of object is in each proposed region (in our case, whether there is a horse in each proposed region or not).

During the first epochs of training, the detector had difficulties detecting any ROIs, but as training progressed, it began making more accurate detections. After 4000 epochs, it was capable of finding the region of interest with high precision.

### 2.3. Model

Next, a model to predict the emotions was created. This model received images of $150 \times 150$ pixels (the rescaled ROI found by the detector) and output predictions (alarmed, annoyed, curious, or relaxed).

The architecture of this model (Figure 2) was formed by a convolutional base, a flattening layer, two fully connected layers (256 and 128 nodes, respectively), and a softmax layer (4 nodes, 1 for each emotion). Three different convolutional bases were tested, the base from ResNet50v2, Xception, and VGG16, all with weights from the imagenet dataset. To train the model, only the last convolutional block of layers and the layers on top of this one were trained. They were trained for 25 epochs using 400 pictures (100 for each category) as the data set and 40 pictures (10 for each category) as the test set. Training the first layers of the base did not make sense in this case. The reason was that these layers learn common patterns that are present in all images, such as corners or straight lines, and since this base was trained using the 14 million images of the imagenet dataset (a popular dataset

to train object recognition models), achieving better performance with only 400 images was unlikely.

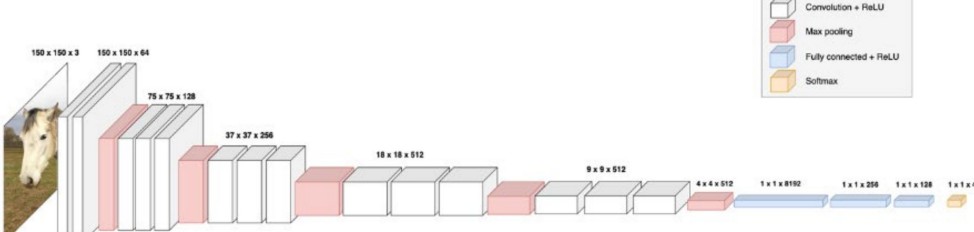

**Figure 2.** Model architecture.

### 2.4. Final Steps

Lastly, in order to facilitate the use of this system, a desktop graphical user interface (GUI) was created. The GUI allows any user to upload an image, processes this image, and displays the ROI with a rectangle (which should be the head and neck of a horse) and the predicted emotion.

In a nutshell, the system created consists of two separate parts, a detector and a model. The detector receives an image previously rescaled to 200 pixels in height and outputs an ROI, a region of the image with a horse. This ROI is rescaled to 150 × 150 pixels and is passed to the model, which predicts and outputs the final emotion. A diagram of the entire process can be seen in Figure 3.

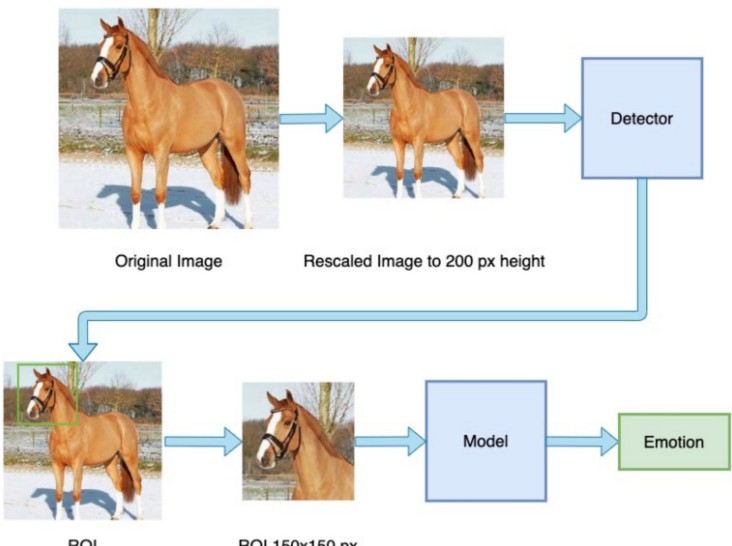

**Figure 3.** System process diagram.

### 3. Results

The work resulted in a detector capable of finding a horse's face in an image, a model capable of predicting the emotion of a horse given a picture of its head, and a user-friendly GUI. All of these partial results are discussed in more detail below.

The detector's precision is very high, and it labeled all 40 validation images without a single error. Some of these detections are shown in Figure 4.

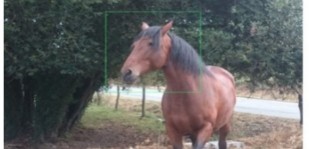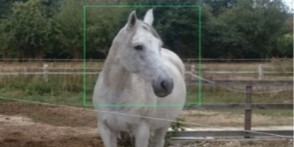

**Figure 4.** Detector predictions.

In order to select the best performing convolutional base, five-fold cross-validation was performed for each of the models. As displayed in Figure 5, the base training set was divided into five equal chunks consisting of 80 images. Using stratified cross-validation, an equal distribution of all four emotion labels was enforced. For each split, a different chunk was selected as the validation set, which results in splits containing 320 training and 80 validation images. After the datasets were formed, a model was trained for each convolutional base and then validated. Thus, each image was tested exactly once and used for training four times. The accuracy of all five iterations was averaged, and the overall performance of a model was calculated. Five-fold cross-validation reduces variance and ensures that the best convolutional base is selected for the final model. Furthermore, there is a risk of overfitting due to the relatively small dataset, and we wanted to ensure the generalization ability of the model by exploiting the full potential of the dataset.

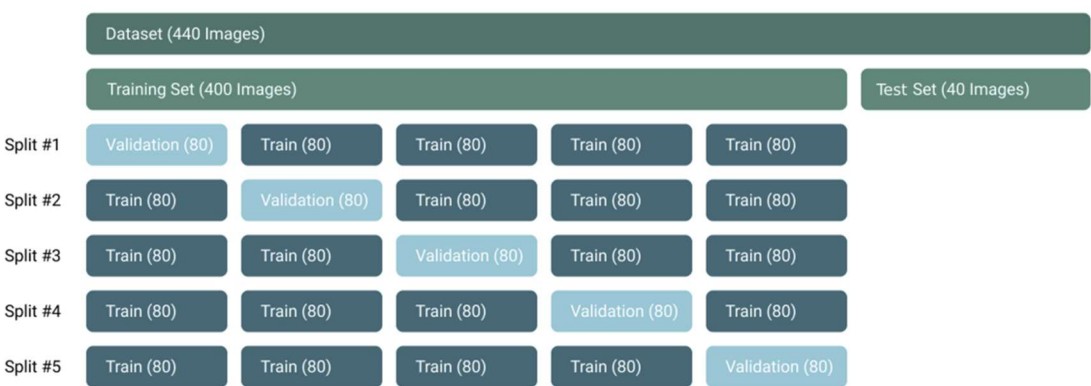

**Figure 5.** Five-fold cross-validation.

Figure 6 displays the average accuracy for each convolutional model base, with accuracy defined as the number of correct predictions divided by the number of total predictions. After training for 25 epochs, the model achieved the best accuracy of 80% using the VGG16 convolutional model.

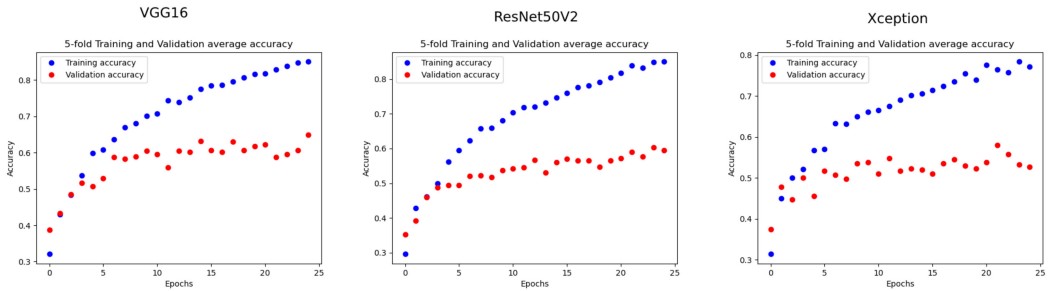

**Figure 6.** Five-fold cross-validation average accuracy over 40 epochs.

In addition, Figures 7 and 8 show the average loss values, i.e., the number of classification errors and the average AUC of each base model. VGG16 has the lowest average validation loss and best AUC value, confirming it as the best performing base model.

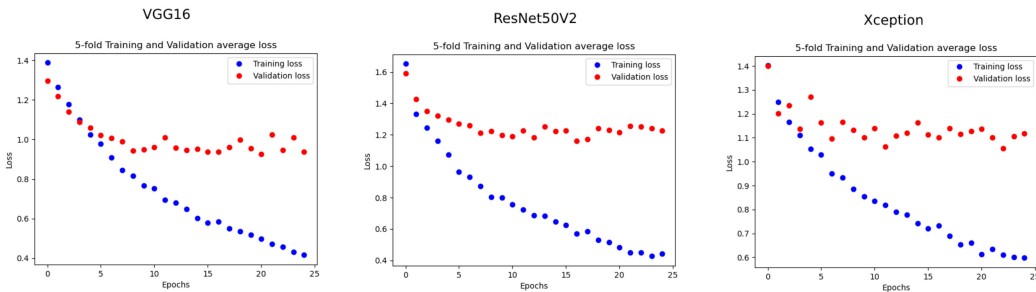

**Figure 7.** Five-fold cross-validation average loss over 40 epochs.

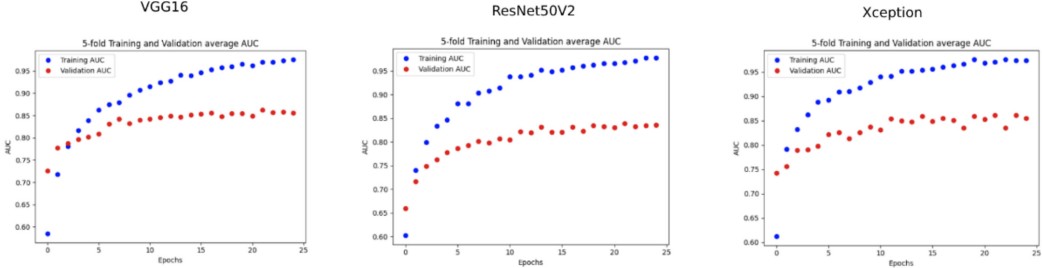

**Figure 8.** Five-fold cross-validation average AUC over 40 epochs.

Based on the VGG16 convolutional base, the model was then trained with the entire training set consisting of 400 images. Testing the model with the remaining 40 images resulted in a validation accuracy of 65%. Figure 9 shows the confusion matrix of this final evaluation.

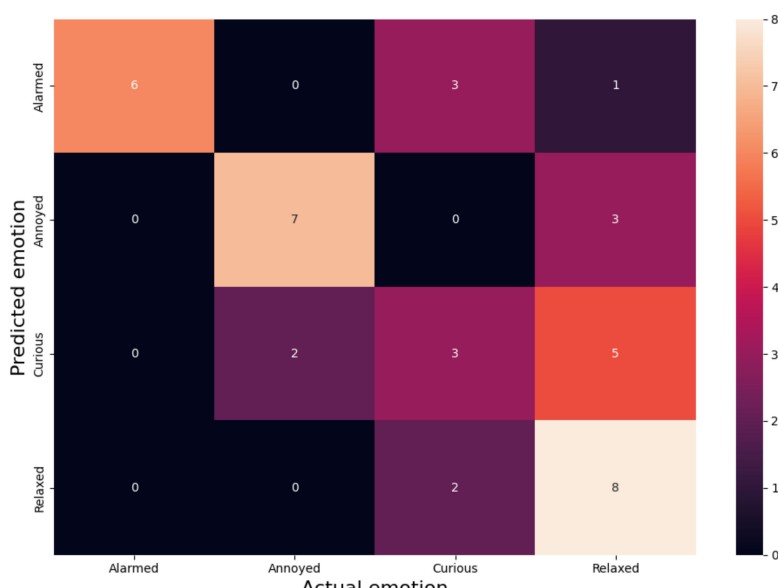

**Figure 9.** Confusion matrix of the final model.

Figure 10 shows some selected test images with a GradCAM overlay, which visualizes activations of the last convolution layer in a heatmap.

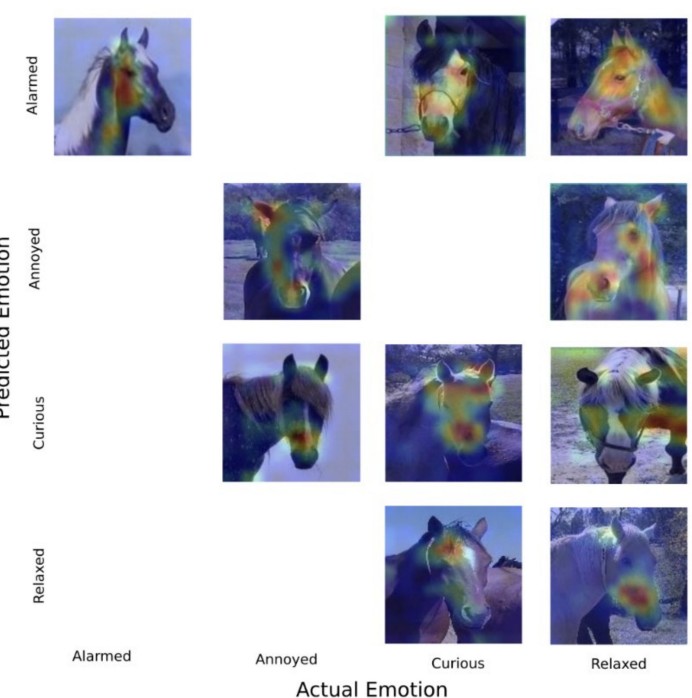

**Figure 10.** GradCAM overlays of the chosen test images.

Finally, the entire system is brought together by the desktop GUI, which includes the two parts and makes them straightforward for anyone to use. This makes it easy for a user to upload an image and see the ROI and the emotion detected. The graphical user interface is shown in Figure 11.

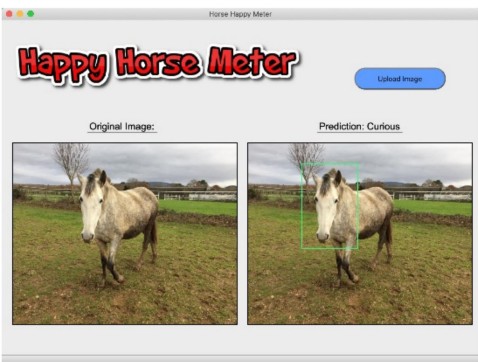

**Figure 11.** Graphical user interface.

## 4. Discussion

This paper describes how an AI-based system capable of detecting emotions of animals was created and able to assess behaviors indicating emotions in horses. The system does so in an autonomous way and with good results. This proves that we are capable of creating a system that can recognize the emotions of a non-human animal species that has the ability to produce facial expressions and that it might be possible to detect these emotions by other methods, such as measuring the animal's heart rate, its temperature, or recording the sounds that they produce and feeding all this data into a system similar to the one created here [41]. While the system works with reasonable accuracy, it is worth pointing out that it could be improved in many ways.

First of all, we have to keep in mind that predicting emotions is a complex task, and it is hard for humans and it is even harder for animals.

Secondly, there were only 440 labeled images in total, which is not a large number for systems like this one. Obtaining the 440 horse emotion pictures was the most time-consuming part of our research, as these pictures were not publicly available and had to be manually taken and labeled by the authors. Future research obtaining more labeled horse emotion pictures will be essential, and we hope to initiate a citizen science initiative towards that goal. Having more images, thousands or millions of them split evenly between every emotion will make future emotion prediction models more accurate and generalizable.

Thirdly, only the head and neck of the horse were used to predict its emotion, and analyzing cues from the entire body would most likely yield better results; however, since we would have more features to analyze, more data would be needed. In addition, if the entire body is used, the emotions have to be reviewed, since they were defined only by cues found in the head and neck of the horse.

Fourthly, ear, neck, and body positions indicative of any general emotion at a distance may also indicate more severe problems upon closer examination. Many of the studies on the equine pain face, for example, show ear and neck positions similar to those of relaxed horses. In order to use such a tool to improve our understanding of equine emotions and promote better equine welfare, this generalized emotional assessment tool could be combined with the existing research and machine learning in equine facial expressions to eventually create a more robust program that takes into account not only the generalized emotional expression, but also the nuances of pain or distress that could be mistaken for something else. This lack of differentiation between subtle emotion patterns is also evident in the confusion matrix in Figure 9, which reveals that emotions are often confused with the "curious" and "relaxed" states of a horse.

Finally, another way to improve the results obtained in this paper would be to use information from other sources in combination with the images. Sensors to measure heart rate and temperature could be put on horses, their sound could be recorded, etc.

With additional adjustments, this tool could serve as an important means of supporting the need to look at animal behavior and emotions as a way to improve animal welfare in various agricultural industries including in clinical veterinary settings [9,10,42]. There is a growing push to create better assessment tools that look at improving equine psychological welfare in domestic states through more robust measurements of emotional affect via behavioral observations [21,43] and an automated system that relies on research-based behavioral assessments rather than objective interpretations could help improve the accuracy of assessment of the psychological welfare of domestic livestock.

## 5. Conclusions

In conclusion, this is a first "proof of concept" system that illustrates that deep learning through convolutional neural networks is able to identify emotions in horses. It provides a powerful foundation on which to build new and more accurate systems to predict animal emotions.

Ethical Statement: This study complies with all ethical regulations and did not involve the use of live animals.

**Author Contributions:** Conceptualization, P.A.G.; Data curation, L.A.C. and E.K.; Formal analysis, L.A.C. and E.K.; Investigation, E.K.; Methodology, E.K. and P.A.G.; Software, L.A.C. and T.S.; Writing—original draft, L.A.C., E.K. and P.A.G.; Writing—review & editing, E.K., T.S. and P.A.G. All authors have read and agreed to the published version of the manuscript.

**Funding:** This research received no external funding.

**Data Availability Statement:** Data available at request from the authors.

**Conflicts of Interest:** The authors declare no conflict of interest.

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
