# Peer review of "Emotion Recognition in Horses with Convolutional Neural Networks"

_futureinternet, doi:10.3390/fi13100250_

Round 1

Reviewer 1 Report

This paper considers a system to recognize and predict the emotions of horses by using convolutional neural networks. The concept of emotion has been analyzed on human beings. However, those for animals are rather interesting. The paper is well organized.

However, I have a few questions which need to be clarified.

  1. The authors defined the four states of emotion from Line 190 to 209.

    I wonder how these states are recognized by image processing techniques.

  1. Do you use any feature for classification?
  2. For the selection of 400 training and 40 validation, how do you label, manual or automatic?
  3. In my opinion, the system process described in Fig 3 is too plain.

It may need a little more details in each block.

  1. The statement from Line 276-278 is rather shallow.

I have not seen any explanation of how these results are obtained.

  1. Fig 4 shows some sample outputs but no sample inputs are shown.
  2. In the paragraph from Line 284 to 295, the training images are 320, and the validation images are 80.

However, in the earlier model, the paper has mentioned 400 training images and 40 validation.

My question is (i) Are 320 from 440? If so how do you make a selection?

It is a little confusing to the readers.

  1. In line 279, it stated that “The detector’s precision is very high, and it labeled all 40 validation images without a single error
  2. But in lines 302-303, it said that after training for 40 epochs, the model achieved the best accuracy of 80% using the VGG16 convolutional model.
  3. Please make a comparison of why it is so.

Apart from these comments, I do not have any objections to the paper.

Reviewer 2 Report

I am honored to have entrusted you with the review of this study.

This study is seen as a paper applying CNN-based image detection and image classificaiton techniques, and it deals with interesting topics, but the following points need to be supplemented.

1. In order to confirm clear labeling for image classification, please show the example image of figure 1 as 5 or more color images for each class.

2. Please show the AUC value for each BackBone network in the result.

3. Please show the confusion matrix in the best prediction model.

4. In order to visually explain the prediction results of the Best prediction model, please use the GradCAM technique to show true positive, true nagative, false positive, false negative image results using 4 or more sample test images.

thank you.

Round 2

Reviewer 2 Report

I am glad that you judged that the quality of the paper submitted by the authors has improved through revision.

In the current state, there is no further inquiry to be made.

This manuscript is a resubmission of an earlier submission. The following is a list of the peer review reports and author responses from that submission.